# Nascent mutant Huntingtin exon 1 chains do not stall on ribosomes during translation but aggregates do recruit machinery involved in ribosome quality control and RNA

**Angelique R. Ormsby, Dezerae Cox[ID], James Daly, David Priest, Elizabeth Hinde, Danny M. Hatters[ID]** *

Department of Biochemistry and Molecular Biology; and Bio21 Molecular Science and Biotechnology Institute, The University of Melbourne, VIC, Australia

* dhatters@unimelb.edu.au

**Data Availability Statement:** All relevant data are within the paper and its Supporting Information files.

## Abstract

Mutations that cause Huntington's Disease involve a polyglutamine (polyQ) sequence expansion beyond 35 repeats in exon 1 of Huntingtin. Intracellular inclusion bodies of mutant Huntingtin protein are a key feature of Huntington's disease brain pathology. We previously showed that in cell culture the formation of inclusions involved the assembly of disordered structures of mHtt exon 1 fragments (Httex1) and they were enriched with translational machinery when first formed. We hypothesized that nascent mutant Httex1 chains co-aggregate during translation by phase separation into liquid-like disordered aggregates and then convert to more rigid, amyloid structures. Here we further examined the mechanisms of inclusion assembly in a human epithelial kidney (AD293) cell culture model. We found mHttex1 did not appear to stall translation of its own nascent chain, or at best was marginal. We also found the inclusions appeared to recruit low levels of RNA but there was no difference in enrichment between early formed and mature inclusions. Proteins involved in translation or ribosome quality control were co-recruited to the inclusions (Ltn1 Rack1) compared to a protein not anticipated to be involved (NACAD), but there was no major specificity of enrichment in the early formed inclusions compared to mature inclusions. Furthermore, we observed co-aggregation with other proteins previously identified in inclusions, including Upf1 and chaperone-like proteins Sgta and Hspb1, which also suppressed aggregation at high co-expression levels. The newly formed inclusions also contained immobile mHttex1 molecules which points to the disordered aggregates being mechanically rigid prior to amyloid formation. Collectively our findings show little evidence that inclusion assembly arises by a discrete clustering of stalled nascent chains and associated quality control machinery. Instead, the machinery appear to be recruited continuously, or secondarily, to the nucleation of inclusion formation.

**Funding:** This work was funded by grants and fellowships from the National Health and Medical Research Council Project to D.M.H. (APP1184166, APP1161803, APP1102059, APP1154352). The funders had no role in study design, data collection and analysis, decision to publish, or preparation of the manuscript.

**Competing interests:** The authors have declared that no competing interests exist.

## Introduction

Huntington Disease (HD) is an incurable and fatal neurodegenerative condition caused by dominant CAG trinucleotide expansion mutations in exon 1 of the *Huntingtin* gene [1]. These mutations expand a polyglutamine (polyQ) sequence in the Huntingtin (Htt) protein to beyond a disease threshold of 35Q, which makes the protein become aggregation prone [2]. Toxicity may also arise from the expanded CAG RNA sequence [3]. N-terminal mutant Htt fragments accumulate in intracellular inclusion bodies (inclusions) during disease progression, which represent a major hallmark of disease pathology [2, 4, 5].

The transgenic expression of the Htt exon 1 fragment (Httex1) in polyQ-expanded form is sufficient to produce a HD-like pathology in rodent and primate models, which is suggestive that these fragments mediate proteotoxicity [6–8]. The mechanism of toxicity remains to be unequivocally determined but it is thought to involve two distinct components; soluble and inclusion states of Httex1 [9]. Soluble states, which may include monomeric or small nanometer-sized oligomers of mutant Httex1 cause oxidative and mitochondrial stress and increase the risk of apoptosis in cell culture models of disease [10–13]. We previously suggested that the toxicity of the soluble forms of mutant Httex1 may involve a quality control feedback mechanism during translation involving stalled Httex1 nascent chains, which when unresolved triggers apoptosis [9]. Once inclusions form survival times are improved in cell culture models of disease, leading to a hypothesis that inclusion formation alleviates toxicity by sequestering the soluble toxic forms away from harm (reviewed in [14]). However, rather than returning the cell to a normal state of homeostasis, cells in culture with inclusions are metabolically quiescent and die at a delayed rate by a non-apoptotic necrotic mechanism [9]. This finding suggests a second level of toxicity from the inclusions distinct to that from the soluble states.

Here we sought to test the hypothesis that newly synthesized mutant Httex1 stalls at the ribosome, misfolds in complex with ribosome associated quality control machinery and aggregates into liquid-like droplets that then over time convert to a rigid structure. Since our initial prediction that mutant Httex1 aggregates may arise through phase separation into liquid-like structures, two studies have since reported evidence supporting this mechanism of action [15, 16]. To avoid confounding influences from repetitive RNA on toxicity, we examined mHttex1 protein encoded with mixed CAG and CAA codons. Our findings suggest that nascent Httex1 does not appear to stall on ribosomes during translation (or that if it does, the levels are minimal) despite an enrichment of machinery involved in ribosome associated quality control into the inclusions. In our hands and model, however, we found the early formed inclusions comprised only immobile mutant Httex1 molecules.

## Methods

### DNA vectors and constructs

Human Httex1 and TC9-tagged Httex1 as fusions to fluorescent proteins were expressed in pT-Rex vectors with CMV-promoters as described previously [9]. The pFN21A-HaloTag constructs were purchased from Promega. The P2A stall construct was prepared as described previously including the preparation of the Httex1 constructs as test sequences [17]. The tandem P2A T2A constructs were made using T2A sequences from [18]. In essence the T2A sequence 5′ GGCGAGGGCAGGGGAAGTCTTCTAACATGCGGGGACGTGGAGGAAAATCCCGGCCCA was inserted after the P2A sequence of the existing Httex1(25Q) sequence in the pTriEx4 vector (GeneArt). The sequence of the derived vector is shown in **Table 1**. From this, other polyQ-length derivatives of Httex1 and the 20K variant was made by excising the gene fragments from the original stall reporter via NotI and BamHI restriction sites. The control linker

**Table 1. Sequence of the base stall construct \*.**

\*The Httex1 construct was inserted into the sequence at position ### with the different polyQ lengths shown below. The other sequences are highlighted as follows. <u>GFP</u>; ***P2A;T2A;*** <u>**mCherry**</u>

CCATGGTGAGCAAGGGCGAGGAGCTGTTCACCGGGGTGGTGCCCATCCTGGTCGAGCTGGACGGCGACGTAAACGGCCACAAGTTCAGCGTGTCCGGCGAGGGCGAGGGCGATGCCACCTACGGCAA
GCTGACCCTGAAGTTCATCTGCACCACCGGCAAGCTGCCCGTGCCCTGGCCCACCCTCGTGACCACCCTGACCTACGGCGTGCAGTGCTTCAGCCGCTACCCCGACCACATGAAGCAGCACGA
CTTCTTCAAGTCCGCCATGCCCGAAGGCTACGTCCAGGAGCGCACCATCTTCTTCAAGGACGACGGCAACTACAAGACCCGCGCCGAGGTGAAGTTCGAGGGCGACACCCTGGTGAACCGCATCGA
GCTGAAGGGCATCGACTTCAAGGAGGACGGCAACATCCTGGGGCACAAGCTGGAGTACAACTACAACAGCCACAACGTCTATATCATGGCCGACAAGCAGAAGAACGGCATCAAGGTGAACTTCAAGA
TCCGCCACAACATCGAGGACGGCAGCGTGCAGCTCGCCGACCACTACCAGCAGAACACCCCCATCGGCGACGGCCCCGTGCTGCTGCCCGACAACCACTACCTGAGCACCCAGTCCGCCCTGAGCAAAGA
CCCCAACGAGAAGCGCGATCACATGGTCCTGCTGGAGTTCGTGACCGCCGCCGGGATCACTCTCGGCATGGACGAGCTGTACAAGGCTAGC***GGAAGCGGAGCTACTAACTTCAGCCCTGCTGAAGCA*** ###
***GGCTGGAGACGTGGAGGAGGAGAACCCTGGACCT***CTGCAGGGCTCCGGCAGGCGGCCGATGAAGCGGCTGGAGAGCGTGGAGGAGGAGAATCCCGGCCCAGCGAGGGCAGGGGAAGTCTTCTAACATGCGGGGACCGTGGA
GGAAAATCCCGGCCCAGAATTCGTGAGCAAGGGCGAGGAGGATAACATGGCCATCATCAAGGAGTTCATGCGCTTCAAGGTGCACATGGAGGGCTCCGTGAACGGCCACGAGTTCGAGATCGAGGGCGA
GGGGCGAGGGCCGCCCCTACGAGGGCACCCAGACCGCCAAGCTGAAGGTGACCAAGGGTGGCCCCCTGCCCTTCGCCTGGGACATCCTGTCCCCTCAGTTCATGTACGGCTCCAAGGCCTACGTGAAGCA
TCTACAAGGTGAAGCTGCGCGGCCACAACTTCCCCTCCGACGGCCCCGTAATGCAGAAGAAGACTATGGGCTGGGAGGCCTGTGATGAACTTCGAGGACGGCGGCGTGGTGACCGTGACCCAGGACTCCTCCCTACAGGACGGCGAGTTCA
GCAGAGGCTGAAGCTGAAGGACGGCGGCCACTACGACGCTGAGGTCAAGACCACCTACAAGGCCAAGAAGCCCGTGCAGCTGCCCGGCGCCTACAACGTCAACATCAAGTTGGACATCACCTCCCACAA
CGAGGACTACACCATCGTGGAACAGTACGAACGCGCCGAGGGCCGCCACTCCACCGGCGGCATGGACGAGCTGTACAAGTAACTCGAG

### sequences shown below

**25Q**
ATGGGCGACCCTGGAAAAGCTGATGAAGGCCTTCGAGTCCCTCAAAAGCTTCCAACAGCAGCAACAGCAGCAACAGCAGCAACAGCAACAA
CCGGCCACCCACCTCCCCCTCCACCCCCCCACCTCCTCAACTTCCTCAACCTCCTCCACCTCCTCCACAGGCCACAGCAGCAACAGCAGCAACAACAGCAGCAACAACAGCAGCAACAGCAGCCGGCCCAGCTGTGGCTGA
GGAGCCTCTGCACCGACCT

**46Q**
ATGGGCGACCCTGGAAAAGCTGATGAAGGCCTTCGAGTCCCTCAAAAGCTTCCAACAGCAGCAACAGCAGCAACAGCAGCAACAGCAGCAACAGCAGCAACAGCAACAA
GCAACAACAGCAGCAACAGCAGCAACAGCAGCAACAGCAGCAACAACCGCCACCCCCCTCCCCCCTCCACCCCCCCACCTCCTCAACTTCCTCAACCTCCTCCACCTCCTCCACAGGCCACA
GCCTCTGCTGCCCGCCACCAACCTCCTCCACCTCCTCCACCTCCTCCACCTCCTCCAGCCACCAGCTGCCACCAACCTCCTCCACCTCCTCCACCTCCTCCACCTCCTCGCACCGACCT

**97Q**
ATGGGCGACCCTGGAAAAGCTGATGAAGGCCTTCGAGTCCCTCAAAAGCTTCCAACAGCAGCAACAGCAGCAACAGCAGCAACAGCAGCAACAGCAGCAACAGCAGCAACAACAGCA
GCAACAGCAGCAACAGCAGCAACAGCAGCAACAGCAGCAACAACAGCAGCAACAGCAGCAACAGCAGCAACAACGCCACCTCCCCCTCCTCCACCTCCTCCACCTCCTCCACCTCCTCAACTTCCTCAA
CAGCAACAACAGCAGCAACAGCAGCAACAGCAACAACCGCCACCTCCTCCACCTCCTCCACCTCCTCCACCCCCCTCCACCCCCCACCTCCTCCACCTCCTCAACTTCCTCAA
CCTCCTCCACCTCCTCCACCTCCTCCACCTCCTCCAGCCACCAGCTGTGGCTGCTGAGGAGCCTCTGCACCGACCT

was made by PCR amplification using forward (5′ GCGGCCGCTATGCCTGGACCTACACC
TAGCG) and reverse (5′ GGATCCGCCGGTTTTCAGGCCAGGGC) primers and ligation into
the P2A T2A Htt25Q Stall Reporter via the NotI and BamHI restriction sites.

## Cell culture

All experiments were performed with the HEK293 cell line derivative AD293, which was main-
tained in DMEM supplemented with 10% (w/v) fetal calf serum (FCS) and 1 mM glutamine in
a 37˚C humidified incubator with 5% v/v atmospheric $CO_2$. For microscopy experiments, cells
were plated at $3 \times 10^4$ cells per well in an 8-well μ-slide (Ibidi). For flow cytometry experi-
ments, cells were plated at $0.5 \times 10^5$ cells in a 24-well plate. Cells were transiently transfected
with the vectors using Lipofectamine 3000 reagent as per manufacturer's instructions (Life
Technologies), and media was changed 6 hours after transfection. For the HaloTag experi-
ments, the transfection was done in a way so as to decouple the correlated expression of the
two plasmids. This involved mixing the plasmids separately with Lipofectamine, before com-
bining the lipofectamine:DNA complexes together to then add to the cells.

## Western blot

AD293 cells were transfected with P2A or P2A T2A stall constructs and harvested 24 hours
after transfection. Cells were pelleted (200 *g*, 6 mins) and resuspended in RIPA lysis buffer
(150mM NaCl, 50 mM Tris pH 8.0, 1% v/v IGEPAL, 0.5% v/v sodium deoxycholate, 0.1% v/v
SDS, 250 U Benzonase and supplemented with cOmplete, EDTA-free Protease Inhibitor Cock-
tail pills (Roche)) and incubated on ice for 30 minutes. Lysate was matched for total protein by
BCA kit (Thermofisher scientific, cat# 23225). 10 μg of total protein lysate was loaded on to an
TGX Stain Free FastCast Acrylamide gel (BioRad, cat# 1610185) and transferred using an
iBlot2 gel transfer device (Thermofisher scientific, cat# IB21001) and a PVDF iBlot2 transfer
stack (Thermofisher scientific, cat# IB24001). The membrane was blocked with 5% w/v skim
milk powder in phosphate buffered saline (PBS) for 1 hour at room temperature. Anti-GFP
(Invitrogen, cat#A6455) and anti-Cherry (Abcam, cat#167453) antibodies were diluted to
1:10,000 and 1:2500 respectively in PBS containing 0.1% v/v Tween 20 and incubated for 1
hour at room temperature. The secondary antibody, goat anti-rabbit HRP antibody (Invitro-
gen, cat#656120), was diluted 1:10,000 in PBS containing 0.1% v/v Tween 20 and incubated for
1 hour at room temperature. HRP was detected by enhanced chemiluminescence.

## FlAsH staining

Cells were stained with FlAsH as described previously to demarcate HBRi from PBRi cells [9].
In essence, the ratio of Cerulean:FlAsH or mCherry:FlAsH fluorescence was determined and
all cells with a ratio greater than one standard error of the mean (SEM) from the mean were
classified as HBRi whereas all cells with a ratio smaller than one SEM from the mean were clas-
sified as PBRi. Note that for different experiments, this calibration strategy led to different
ratio threshold values. Hence for each experiment with reported ratios, we also report the
mean and SEM values that were measured.

## RNA staining

Cells were stained for RNA using the Click-iT Plus Alexa Fluor647 Picolyl Azide Toolkit (Life
technologies, cat#C10643) according to manufacturer's instructions. In short, 6 hours post-
transfection 5-ethynyl uridine (EU) (Click-Chemistry tools, cat# 1261–10) was added to cells
to a final concentration of 0.4 mM. 24 hours post-transfection cells were stained with FlAsH as

described above. Following staining, cells were fixed with 4% w/v paraformaldehyde and permeabilised with 0.5% w/v Triton X-100 in PBS. Cells were then washed with 3% w/v bovine serum albumin in PBS followed by Click-It staining according to manufacturer's instructions using a 1:4 ratio of $CuSO_4$: Copper protectant. Nuclei were counter stained with Hoechst (1:1000) and imaged by confocal microscopy.

## HaloTag staining

After 6 hours post-transfection with the Halo-tagged constructs, TMRDirect ligand (Promega) was diluted 1:1000 in complete media and added to cells. For imaging, cells were fixed at 24 hours post-transfection with 4% w/v paraformaldehyde. Imaging positions were marked using the "mark and find" feature and then images captured on a Lecia TCS SP5. Cells were then stained with FlAsH as described above and the positions were re-imaged.

## Flow cytometry

Cells were analyzed at high flow rate in an LSRFortessa flow cytometer, equipped with 488- and 561-nm lasers (BD Biosciences). 50,000–100,000 events were collected, using a forward scatter threshold of 5,000. Data were collected in pulse height, area, and width parameters for each channel. For Cerulean fluorescence, data were collected with the 405-nm laser and BV421 filter (450/50 nm). TMR fluorescence was collected with the 561-nm laser and using the PE-Texas Red filter (610/20 nm). All flow cytometry data were analysed with FlowJo (Tree Star Inc.). First, live cells were gated using forward and side scatter parameters and cells with inclusions were further gated from cells without inclusions by Pulse Shape Analysis [19]. The fluorescence intensity for individual cells in each channel was exported and analysed using custom python scripts (available at https://doi.org/10.5281/zenodo.3789864). Briefly, cerulean fluorescence was assigned to 20 logarithmic bins spanning the range of recorded intensities. Four logarithmic bins for HaloTag fluorescence were then assigned (none, low, medium, high) independently for each construct according to the maximum intensity for that construct. Finally, the proportion of cells containing inclusions within each combined cerulean-HaloTag bin was calculated.

## Immunofluorescence

Cells were fixed at 24 hours after transfection with 4% w/v paraformaldehyde for 15 min at room temperature. Cells were then permeabilized with 0.2% v/v Triton X-100 in PBS for 20 mins at room temperature. Samples were blocked in 5% w/v bovine serum albumin in PBS for 1 hour at room temperature. Cells were stained with anti-GFP (1:300 dilution) (Invitrogen cat# A6455) diluted in PBS containing 1% w/v bovine serum albumin and 0.05% v/v Tween 20 for 1 hr at room temperature. Samples were then incubated in goat anti-rabbit Alexa Fluro 647 (1:500) (Life technologies cat#A21244) diluted in PBS containing 1% w/v bovine serum albumin and 0.05% v/v Tween 20 for 30 mins at room temperature.

## Fluorescence recovery after photobleaching

Cells were imaged at 37˚C and 5% atmospheric $CO_2$ on an Olympus FV3000 confocal laser scanning microscope through a 60X 1.2NA water immersion objective. mCherry fluorescence was excited with a 561 nm solid state laser diode. The resulting fluorescent emission was directed through a 405/488/561 dichroic mirror and detected by an internal GaAsP photomultiplier set to collect between 550–650 nm. 24 hours after transfection cells were FlAsH stained as described above and then imaged. For FRAP, a pre-bleached image was taken before half

the inclusion was bleached for 5 seconds (10% laser power, 200 µs pixel dwell time). Recovery was then monitored by imaging the inclusions every minute for 21 minutes.

## Cell imaging and analysis

Images of live or fixed cells were acquired with a Leica TCS SP5 confocal microscope. Anti-GFP images were taken on the Zeiss LSM800 Airyscan. Images were extracted from proprietary formats using FIJI (v 2.0.0-rc-69/1.52p) equipped with the Bioformats plugin (v6.3.1). The resultant tiff images were then processed using custom scripts written for FIJI and python (source code available at https://doi.org/10.5281/zenodo.3789864). Briefly, in the case of RNA- and HaloTag-stained images, regions of interest (ROI) were manually assigned and the mean fluorescence intensity within each region exported. In the case of fluorescence recovery after photobleaching, bleached and whole-cell ROI were assigned via automatic thresholding of the FlAsH and mCherry channels respectively, while a circular background ROI with a nominal fixed radius (25 pixels in the images) was manually assigned for each image. Pixel fluorescence intensities for each ROI were exported for all timepoints post-recovery, and the bleached ROI determined by subtracting the unbleached pixel coordinates from the whole-cell ROI. The relative recovery was then calculated by dividing the background-corrected mean intensity in the bleached and non-bleached ROIs. Finally, in the case of anti-GFP antibody penetration, inclusions were identified by automatic thresholding of the cerulean intensity and the resultant ROI was scaled by 110% to yield the external inclusion boundary. The internal boundary was determined via automatic thresholding of the inverted anti-GFP fluorescence. The Euclidean distance between the centroid of the outer ROI and either the internal or external boundary pixels was then calculated, and the penetration measured as the difference in the mean distance of the internal and external boundaries.

## Statistics

The statistical tests are described in the figure legends and $P$ values shown directly on the figures or coded as [*], $P < 0.05$, [**]; $P < 0.01$, [***]; $P < 0.001$; [****], $P < 0.0001$. Statistical tests were performed in Graphpad Prism v6.

## Results

We previously postulated that in cells lacking Httex1 inclusions, soluble mutant Httex1 was recognized as abnormal by an unknown translation-related quality control mechanism. Evidence for this mechanism comes from soluble polyglutamine (polyQ)-expanded Httex1 being more efficiently degraded than the wild-type counterpart [20]. One hypothesis to explain this result is that nascent mutant Httex1 stalls on the ribosome during synthesis, which triggers a ribosome quality control clearance response. Our previous work found Upf1 (Rent1), which plays a central role in non-sense mediated decay [21] as being enriched in the inclusions [9]. This finding raises the possibility that the stalled constructs, should they arise, proceed to nucleate the inclusion assembly process.

First to test for stalling, we implemented a translational stall assay in AD293 cells, which are sensitive to the proteotoxicity of soluble polyQ-expanded Httex1 [9]. The assay involves a reporter cassette containing two fluorescent reporters on each side of the peptide sequence to be tested for stalling (GFP at the N-terminus and mCherry at the C-terminus) [22] (**Fig 1A**). Each construct is encoded in frame without stop codons however the test sequence is flanked by viral P2A sequences, which causes the ribosome to skip the formation of a peptide bond but otherwise continue translation elongation uninterrupted [23]. Complete translation of the cassette from one ribosome will generate three independent proteins (GFP, test protein, and mCherry). However, should the ribosome stall during synthesis (such as through the

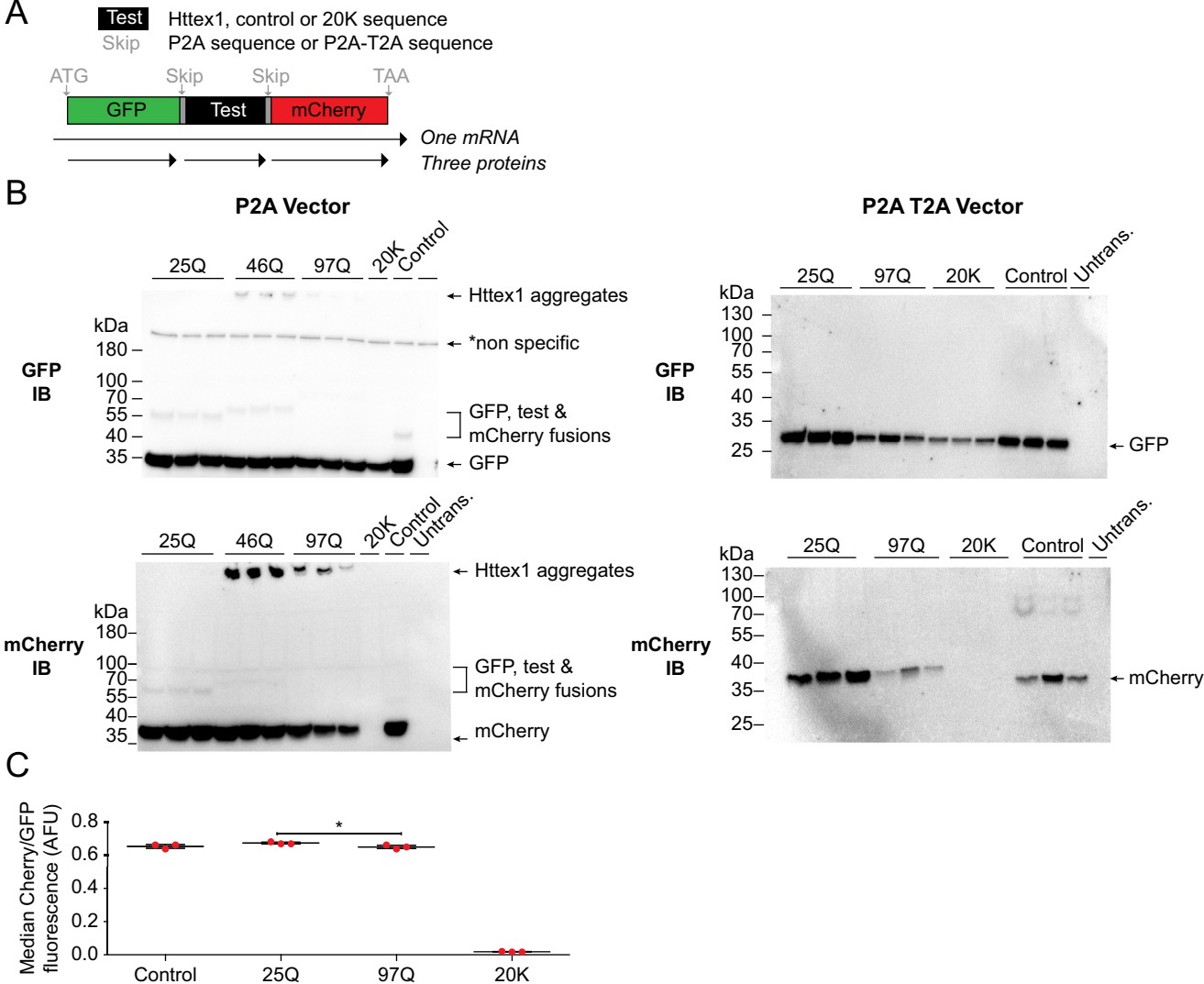

**Fig 1. Nascent chains of Httex1 are not stalled during synthesis with short or long polyQ lengths in AD293 cells. A.** Schematic of the design of the stall reporter assays. GFP and mCherry sequences flank a test sequence and are expressed in frame. The 20K sequence is a positive stall control containing repeat AAA sequences as the test sequence. The Control linker is a non-stalling sequence for the test sequence. In the original assay, the P2A sequence causes the ribosome to skip the formation of a peptide bond leading to independent protein expression of the three components. If the ribosome stalls during translation of the test sequence, then the yield of mCherry decreases relative to GFP. In the modified assay, an additional alternative skip sequence (T2A) was added in tandem to the P2A to reduce the rate of readthrough **B.** Western blots of cells expressing the stall reporters probed with GFP or mCherry antibodies. Marker molecular weight masses are shown on the left of each blot. Products labeled on the right of each blot. **C.** mCherry:GFP fluorescence ratios measured by flow cytometry.

previously established poly-lysine (20K) sequence used here as a control [22]), mCherry is produced at lower stoichiometries than the GFP.

Examination of various test sequences revealed free mCherry and GFP protein products as anticipated (**Fig 1B**). However, we also observed higher molecular mass entities that are of the predicted size for P2A sequence readthrough products to create fusion proteins (**Fig 1B**). Of particular note was the appearance of GFP and mCherry immunoreactive SDS-insoluble material in the stacking gel when we used mutant Httex1 (46Q or 97Q) as the test protein, which is indicative of SDS-insoluble mHtt fusion products that arises from aggregation [4] (**Fig 1B**). Because fluorescence resonance energy transfer (FRET) from GFP to mCherry is anticipated

to inflate the mCherry/GFP fluorescence ratio, and be particularly high in the aggregates, we sought to remove this confounding factor by redesigning the stall construct to reduce the rate of readthrough [18]. This involved inserting an additional skip sequence (T2A) after the P2A sequence (**Fig 1A**). The resultant constructs yielded masses consistent with the absence of any read through products (**Fig 1B**). Analysis of the fluorescence ratios of mCherry:GFP by flow cytometry, which reports on levels of stalling, indicated no major difference between the wild-type (25Q) and mutant (97Q) Httex1 forms compared to the control construct, although there was a very small significant difference between the 25Q and 97Q constructs (**Fig 1C**). Collectively these data suggest that long polyQ sequences at best lead to very small levels of stalling and therefore points to nascent chain stalling as unlikely to be a major factor for the proteo-toxicity of soluble mHttex1 or in driving the formation of inclusions.

Next, we assessed whether the inclusions contained embedded RNA, which may arise if nascent chains co-aggregate with ribosomes during translation into the inclusions. For this we used a biosensor form of Httex1 (TC9-Httex1) we previously developed that enables us to demarcate early-formed inclusions from older inclusions [9, 19]. This biosensor involves a tet-racysteine tag embedded near the start of the polyglutamine sequence that can bind to the biar-senical dye FlAsH when Httex1 forms disordered aggregates that we previously showed were enriched in early-formed inclusions [9]. In contrast, TC9-Httex1 loses the capacity to bind FlAsH when Httex1 is in an amyloid conformation, which is enriched in more mature inclu-sions [9]. Thus, by examining TC9-Httex1 fused to a fluorescent protein (Cerulean), the ratio of FlAsH reactivity to Cerulean fluorescence gives an indication to whether an inclusion is newly formed or more mature. We had previously developed a flow cytometry method called Pulse Shape analysis to gate out cells with early-formed inclusions from more mature inclu-sions [19, 24]. In essence, cells with inclusions (i) can be gated from cells with only soluble Httex1 (ni) and then further gated into those Highly Biarsenical-Reactive (HBR) or Poorly Biarsenical-Reactive (PBR) [19, 24]. The result is the detection of cells with recently formed inclusions that are reactive to FlAsH (HBRi), and cells with mature inclusions that are PBRi [9]. HBRi and PBRi status can also be determined by a microscopy approach whereby ratios of FlAsH:Cerulean fluorescence inside the inclusions correlate to PBR status (low ratios) and HBRi status (high ratios) [9]. For each experiment described here, the absolute ratio values are relative to that experiment. HBRi are defined as those inclusions higher than the SEM of the mean and PBRi those that are lower than the SEM of the mean.

To test for incorporation of RNA we used a Click-It RNA imaging approach, which involves adding 5-ethynyl uridine (EU) nucleoside to the media that then gets incorporated into newly synthesized RNA molecules and can be labelled with a fluorophore by click chemis-try [25]. EU was rapidly incorporated into nuclear RNA pools and to a lesser extent the cyto-plasm (**Fig 2A**). The levels of EU inside the inclusions was lower than the surrounding cytosol (as assessed by microscopic imaging) but was higher than background levels of fluorescence determined by control cells labelled with Alexa 647 in the absence of EU, suggesting that a small level of mRNA gets incorporated into the core structure of the inclusions (**Fig 2B**). How-ever the levels of EU in the inclusions did not correlate with FLAsH:Cerulean ratios, which suggested there was no difference in abundance of RNA inside the early-formed inclusions (HBRi) compared to more mature (PBRi) inclusions (**Fig 2B**). There was also a negative corre-lation between FlAsH:Cerulean ratios with cytoplasmic EU staining (**Fig 2B**), which we inter-pret as resulting from higher expression levels (and hence mRNA) of Httex1 in the cells with the lower FlAsH:Cerulean ratios. This is consistent with our previous findings that PBRi status correlated with higher expression levels of Httex1 [9]. The data here collectively suggested that mRNA incorporation into inclusions does appear to occur, however this may not be specific to the first steps of inclusion assembly.

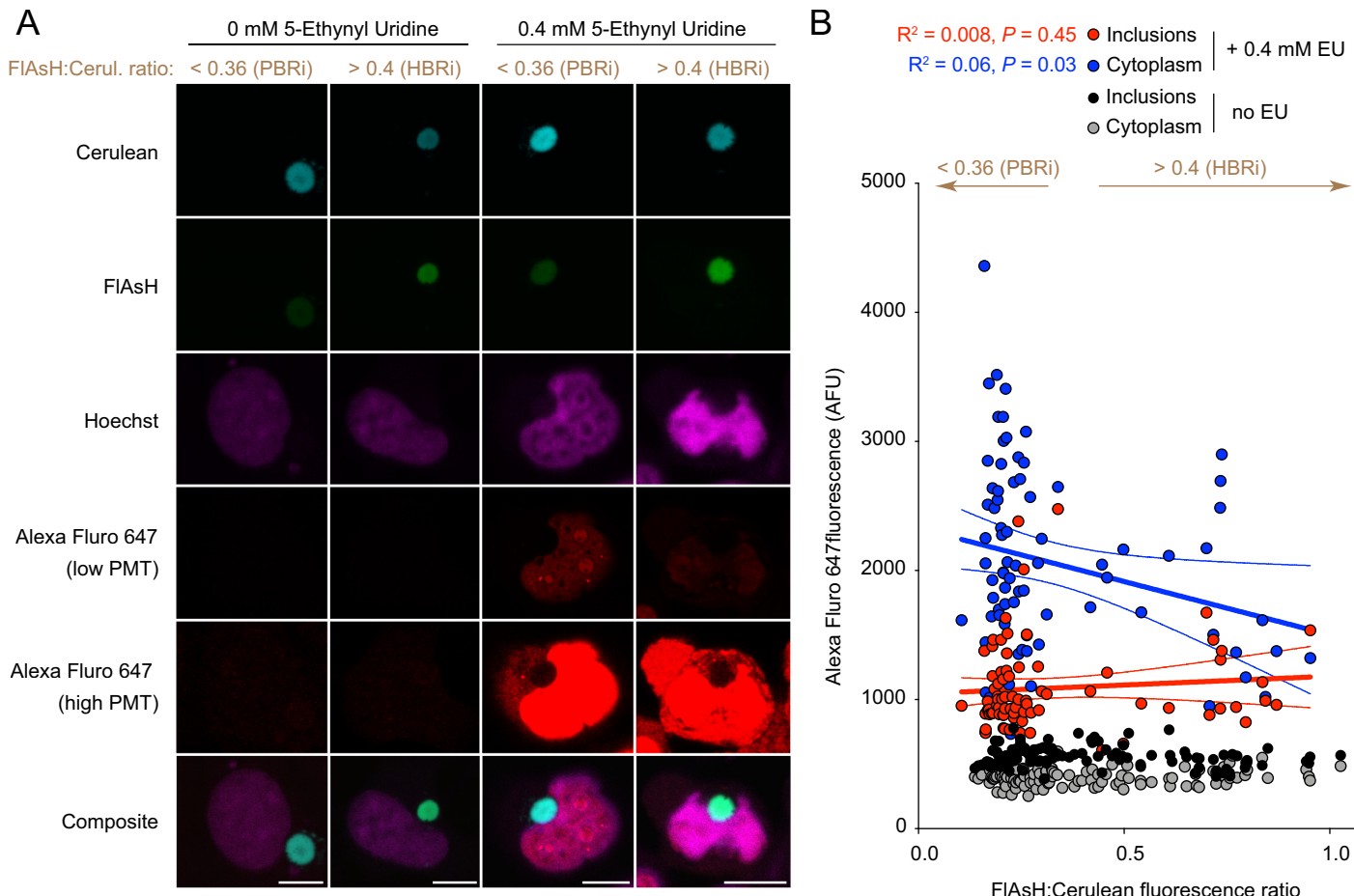

**Fig 2. RNA appears embedded inside inclusions, but is not enriched in the early formed Httex1 inclusions compared to mature inclusions. A**. Confocal images of AD293 cells transfected with TC9-Httex1(97Q)-Cerulean and stained with FlAsH 16 h after growing in nucleoside analogue 5-ethynyl uridine. Cells were stained with Hoecsht 33342 (nuclei stain), and Alexafluor 647 (RNA). Scale bar: 10 μm. **B.** Quantitation of individual images of cells for intensity of Alexafluor 647 in one focal plane for cells supplemented with 0.4 mM EU or no EU as a control (measured by confocal microscopy). Linear regression analyses is shown (thick lines the best fit, and narrow flanking lines the 95% confidence intervals). Arrows (brown lines and text) represent the classification for HBRi and PBRi based on FlAsH:Cerulean ratios higher and lower than the SEM from the mean (mean of 0.379 and SEM of 0.017).

For our next set of experiments, we sought to test whether proteins potentially engaging with newly synthesized Httex1 were preferentially recruited to the early formed inclusions, and whether their presence influenced the formation process of inclusions. First we examined components of the 60S ribosome and other candidate machinery involved in ribosome quality control (RQC). The RQC system is responsible for monitoring partially synthesised proteins and labelling nascent chains that stall before reaching the stop codon for destruction [26, 27]. The RQC forms a stable complex with the 60S ribosome which triggers degradation of the nascent chain via ubiquitinylation [27]. The sequence of events include (1) splitting of the stalled ribosome; (2) assembly of the RQC and ubiquitinylation of the nascent chain; (3) extraction of the nascent chain and then degradation [26].

For these experiments, we co-expressed Halo-tagged candidate proteins previously suggested to be involved in ribosome quality control with TC9-Httex1(97Q), allowed inclusions to form and then stained for location of the candidate proteins via the HaloTags. Early and mature inclusions were identified using FlAsH:Cerulean fluorescence ratios. This included

Rack1 (Gnb2l1), which is a component of the 40S ribosome and is involved in translational repression [28, 29]. Rack1 has previously been reported to bind to Httex1, which in mutant form was suggested to interfere with protein translation [30], and is involved in initiating RQC by promoting ubiquitinylation of 40S ribosome subunits to resolve poly(A)-induced ribosome stalling [31]. We examined listerin (Ltn1) which is an E3 ubiquitin ligase that is recruited to 60S ribosome subunits close to the nascent chain and ubiquitinylates nascent chains that become stalled during synthesis [27]. Rack1 was relatively evenly distributed throughout the core structure of the inclusion for HBRi and PBRi (**Fig 3A**). By contrast, Ltn1 was enriched to the outer shell of both HBR1 and PBRi inclusions, suggesting it is not a component of the initial inclusion nucleus (**Fig 3B**). We also investigated another protein not observed as enriched

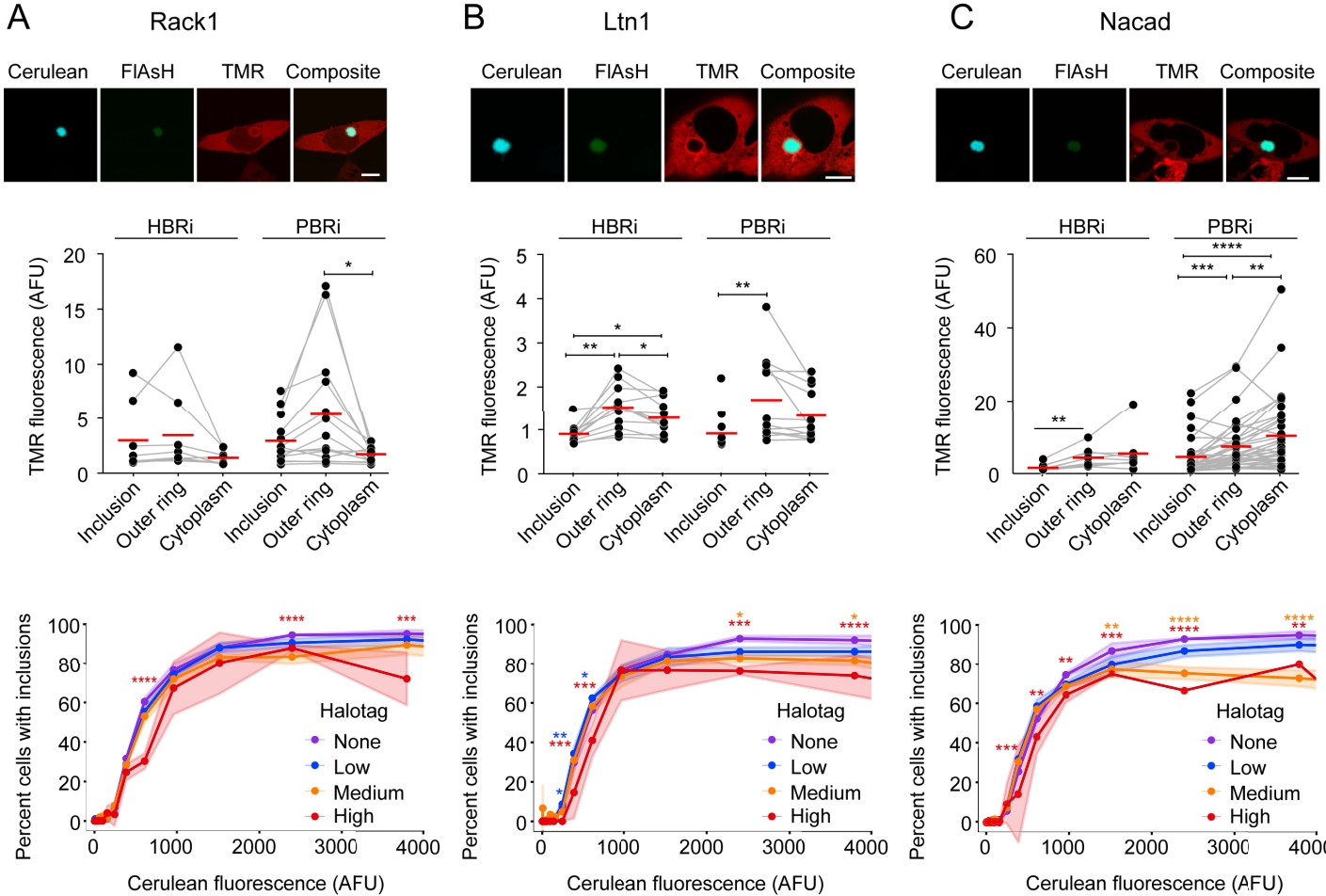

**Fig 3. RQC proteins are recruited and enrich in the outer layer of early-formed and mature inclusions whilst having no affect on inclusion formation.** AD293 cells were co-transfected with TC9-Httex1(97Q)-Cerulean and Halo-tagged RQC proteins Rack1 (**A**), Ltn1 (**B**) and Nacad (**C**). HaloTag proteins were visualized by staining with TMRDirect ligand 24 h after transfection (scale bar; 10 μm). Images (*upper panels*) show cells with inclusions classified as PBRi (with FlAsH:Cerulean ratio of 0.049 (mean = 0.063, SEM = 0.002), 0.106 (mean = 0.115, SEM = 0.002) and 0.112 (mean = 0.180, SEM = 0.020) respectively. Enrichment of Halo-tagged proteins in inclusions were then quantified (*middle panels*). The outer ring denotes the outer edge of the inclusion which was defined as the outer most area of cerulean fluorescence in the inclusions. Data points indicate individual cells, with lines connecting matched data. Means are shown as red dashes. Differences were assessed by repeated measures one way ANOVA with Tukey multiple comparisons test. Impact of HaloTag protein expression on Httex1(97Q)-Cerulean inclusion formation was determined by flow cytometry analysis (*lower panels*). Data indicates fraction of cells with inclusions, as measured by Pulse Shape analysis for cells categorized into different expression level bins (Htt levels, Cerulean fluorescence and Halo-tagged protein level, TMR fluorescence). Cerulean fluorescence is a measure of the abundance of the Httex1 in the cell and for the flow cytometry we did not label the cells with FlAsH. The TMR was categorized into four log-spaced expression-level bins for each protein. Points and lines show means, shaded regions indicate SD. Significance was calculated with a two-way ANOVA with Dunnett multiple comparisions test.

in inclusions [9] or predicted to be involved in this biology as a negative control, the α domain-containing protein 1 (Nacad). Nacad did not colocalize with the inclusions based on HaloTag staining, providing confidence that the enrichment of Rack1 and Ltn1 to inclusions is not an artefact (**Fig 3C**). The over-expression of these proteins did not appear to have large effects on the formation of inclusions of Httex1(97Q), even though there were statistically significant differences (**Fig 3A–3C**). Collectively these data point to Rack1 and Ltn1 as being aggregated into the inclusions but neither appeared to be enriched more specifically in the core of the early formed inclusions, suggesting they are not specifically participating in the nucleation of inclusion formation.

To further probe whether there is sequential recruitment of other proteins previously shown to be enriched in Httex1(97Q) inclusions [9] into early-formed and mature inclusions, we overexpressed Halo-tagged versions of proteins most enriched proteins in PBRi inclusions (Hspb1, Sgta, Upf1) and HBRi inclusions (Snu13, Rpl18) (**Fig 4A–4D**). Hspb1 (Hsp27), which is a small heat shock protein involved in chaperone activity, and Upf1 were enriched in the outer shell of HBRi inclusions (**Fig 4A and 4D**). Hspb1 became more enriched in the outer shell of PBRi inclusions whereas Upf1 remained more evenly distributed through the core structure of the PBRi inclusions. Sgta, which is a co-chaperone involved in the Bag6 system and ERAD and Snu13, which is involved in pre-mRNA splicing, were also enriched more evenly throughout the HBRi inclusion structure but became more relatively enriched to the outer shell of the PBRi inclusions (**Fig 4B and 4C**). These patterns suggested that Hspb1, Sgta and Snu13 were recruited progressively as inclusions formed, grew, and matured. All three proteins suppressed the aggregation of Httex1(97Q) at high co-expression levels, with Hspb1 and Snu13 having a more potent effect (**Fig 4A–4C**). Previously we found Rpl18 as the most enriched protein in Httex1 inclusions by proteomics [9]. Halo-tagged Rpl18 was mostly present in the nucleus as punctate structures but small levels were seen in the cytosol (**Fig 4E**). Endogenous Rpl18 is anticipated to mostly reside in the nucleolus, ER and cytoplasm based on the Human Protein Atlas database [32]. We did not observe any evidence of enrichment of Halo-tagged Rpl18 with Httex1 inclusions (HBRi or PBRi). Furthermore, there was no evidence that the expression level of Halo-tagged Rpl18 affected the aggregation of Httex1 into inclusions (**Fig 4E**). However, the strong nuclear localization of the Halo-tagged Rpl18 suggests it might not be properly forming complexes with the ribosomes under these conditions.

Our last set of experiments examined the diffusibility of mHttex1 molecules inside the HBRi and PBRi inclusions to test for molecular rigidity or liquidity [9, 15, 16]. For this experiment we performed fluorescence recovery after photobleaching on live AD293 cells expressing the TC9- Httex1(97Q)-mCherry and stained with FlAsH. One hemisphere of each inclusion was bleached (as indicated in the **Fig 5A and 5B**) and monitored for recovery of fluorescence in the bleached region over 20 mins. No recovery was observed regardless of whether the inclusions were classified into HBRi and PBRi over a period of 20 mins, which is consistent with a non-liquid aggregation state (**Fig 5A** and **5B**). Hence these data indicate the early formed inclusions comprised Httex1 molecules in an immobile and non-diffusible state and thus the Httex1 molecules become rigid soon after the inclusion process begins.

To further probe for structural differences between the early-formed and mature inclusions we used an antibody permeation assay as a probe for inclusion porosity. Cells expressing TC9-Httex1(97Q) fused to Cerulean were first stained with FlAsH and then immunostained with a GFP antibody. The inclusions were brightly stained at the surface, with no correlation in antibody penetration with FlAsH/Cerulean ratio, suggesting that early formed inclusions have a similar porosity to the mature inclusions (**Fig 5C**). The results here collectively suggested that inclusions form a dense, and immobile core structure quickly after formation (**Fig 5D**).

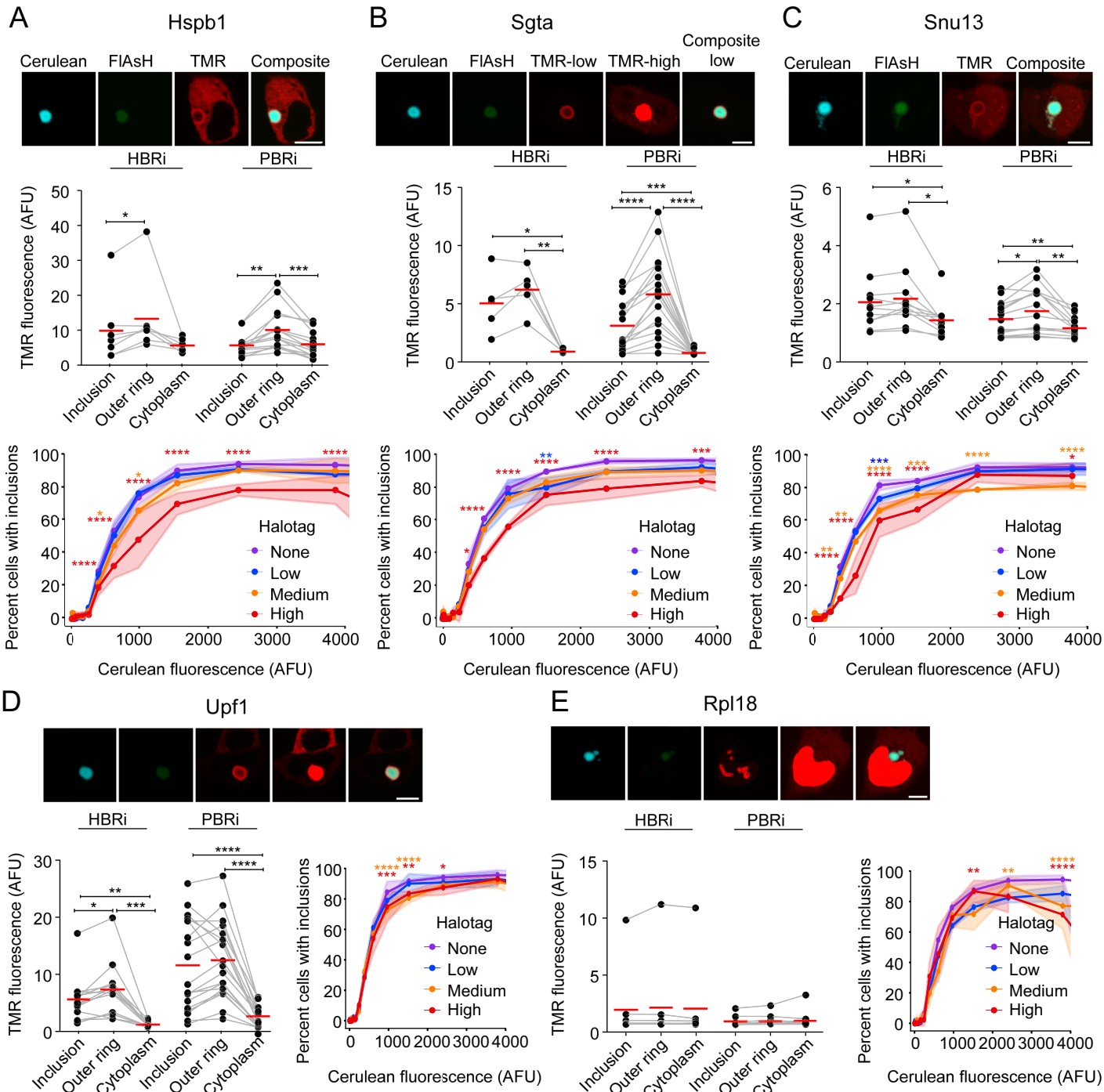

**Fig 4. Recruitment patterns of proteins previously shown to be enriched in Httex1(97Q) inclusions and impact on inclusion assembly.** AD293 cells were co-transfected with TC9-Httex1(97Q)-Cerulean and either chaperones Hspb1 (**A**) or Sgta (**B**), splicing proteins Snu13 (**C**) or Upf1 (**D**), or ribosome protein Rpl18 (**E**). Data is presented similarly to Fig 3. Images (*upper panels)* show representative cells classified as PBRi inclusions (with FlAsH:Cerulean ratios of 0.047 (mean = 0.0547, SEM = 0.002), 0.099 (mean = 0.138, SEM = 0.013), 0.099 (mean = 0.140, SEM = 0.004), 0.135 (mean = 0.222, SEM = 0.011) and 0.120 (mean = 0.169, SEM = 0.003) respectively. Scale bar; 10 μm. For proteins where cytosolic TMR staining could not be seen when other structures were not saturated, a second 'high PMT' image was taken where the PMT was increased to a point when the cytoplasm could then be seen. Differences in HaloTag localisation (*middle panels for A-C and lower left for D and E*) were assessed by repeated measures one way ANOVA with Tukey multiple comparisons test. Impact of HaloTag protein expression on Httex1(97Q)-Cerulean inclusion formation was determined by flow cytometry analysis (*lower panels for panels A-C and lower right panels for D and E*). Data indicates fraction of cells with inclusions, as measured by Pulse Shape analysis for cells categorized into different expression level bins (Htt levels, Cerulean fluorescence and Halo-tagged protein level, TMR fluorescence). Cerulean fluorescence is a measure of the abundance of the Httex1 in the cell and for the flow cytometry we did not label the cells with FlAsH. The

TMR was categorized into four log-spaced expression-level bins for each protein. Points and lines show means, shaded regions indicate SD. Significance was calculated with a two-way ANOVA with Dunnett multiple comparisions test.

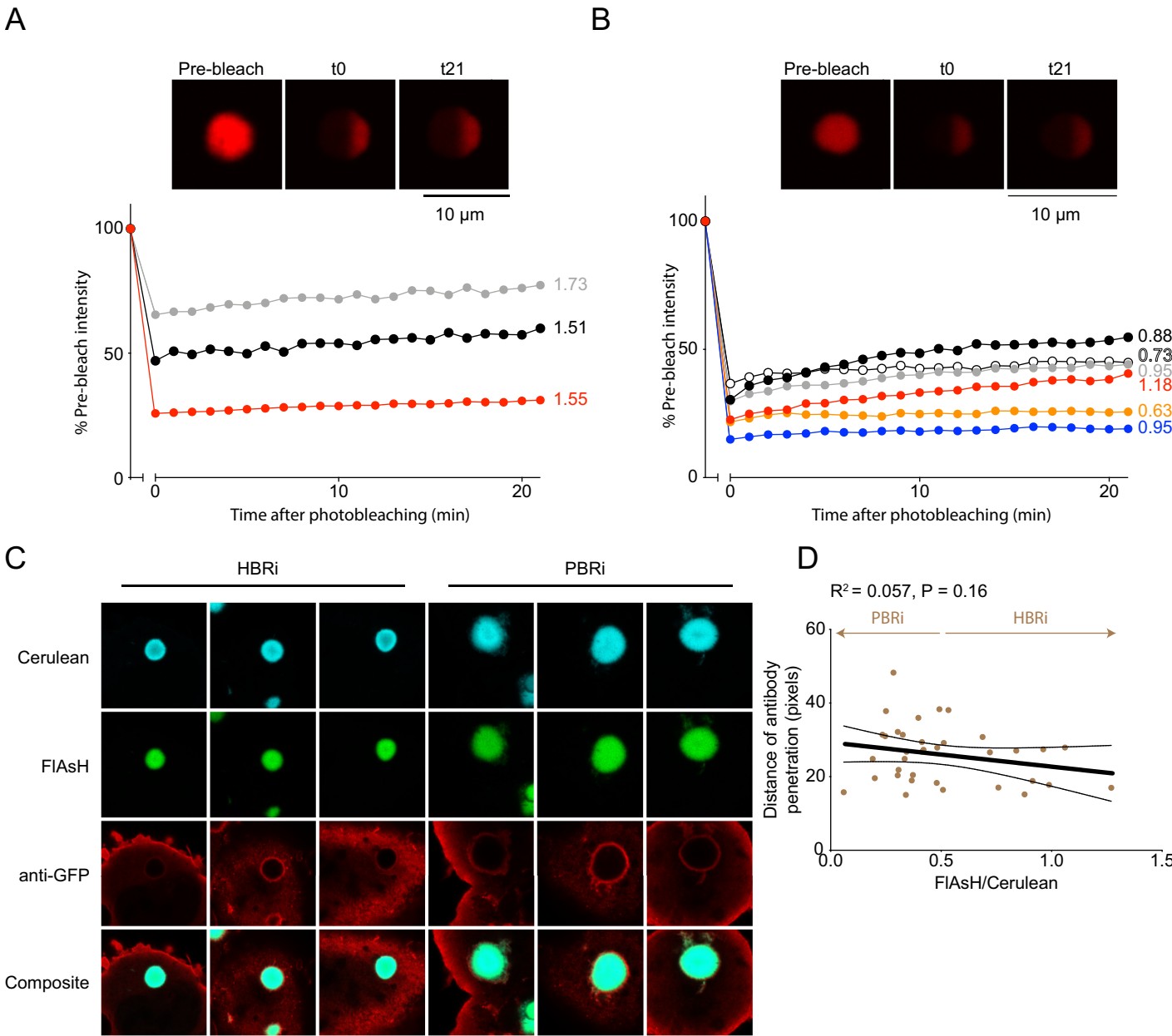

**Fig 5. Httex1 molecules are immobile within early and later-formed inclusions.** Fluorescence recovery after photobleaching experiments tracking the recovery of mCherry fluorescence in regions of bleached HBRi (**A**) and PBRi (**B**) inclusions. Cells were expressing TC9-tagged Httex1 as a fusion to mCherry. Shown are representative images of inclusions before (pre-bleached), immediately after bleaching (t0) and at the endpoint (t21) following recovery. Plots show fluorescence recovery in the bleached region for individual inclusions (each inclusion is one colour). The numbers refer to the FlAsH/Cerulean ratio used to assign inclusion status as HBRi and PBRi (with population mean of 0.504 and SEM of 0.047). **C.** Inclusions in cells expressing Httex1 fusions to Cerulean are impervious to anti-GFP antibodies. Shown are representative images of anti-GFP stained inclusions for HBRi and PBRi (scale bar: 10 μm) **D.** Quantitation of antibody penetration distance into inclusions (pixels) from confocal images of immunostained inclusions in situ in cells. Individual inclusions are shown with the FlAsH:Cerulean ratios inside the inclusions. Shown are the classification values for HBRi and PBRi status (based on mean of 0.330 and SEM of 0.026). Linear regression analysis is shown (thick lines show the line of best fit whilst the narrow flanking lines show the 95% confidence interval). There was no statistical difference as assessed by a two tailed t-test ($P$ = 0.1641).

## Discussion

Our data here suggest that expression of Httex1, at wild-type or mutant polyQ lengths, does not stall on the ribosome during translation or does so very modestly in the best-case scenario. Furthermore, the proteins that are involved in ribosome quality control did not appear specifically enriched in the early formed inclusions relative to the mature inclusions. Because we saw evidence of RNA in the inclusions, these data point to one of two likely scenarios. One is that all these effects are collateral participants to the aggregation process or are secondary in the sequence of events to other more specific mechanisms driving inclusion formation. The other scenario is that nascent chains and ribosomes are recruited to the inclusions and that their contribution to inclusion assembly happens continuously as inclusions grow and mature.

At the time of writing a study was published that suggested Htt has a physiological function of binding to ribosomes and slowing the speed of ribosome translocation of many target mRNAs [33]. Intriguingly, mutant Htt further slowed protein synthesis rates [33]. These data suggest that Httex1 may be slowing translation in trans rather than in cis (through nascent Httex1-driven stalling). This data therefore raises the possibility that mutant Httex1 does not itself stall during synthesis but upon aggregation can sequester proteins similarly involved in regulating ribosome translocation, and potentially those involved in ribosome quality control, into mutant Httex1 inclusions. This mechanism likely would involve the co-aggregation of Httex1 with the endogenous full length Htt that exerts the physiological activity. It is important to note that our constructs contained mixed CAG and CAA codons of glutamine, which is different to the mostly homogenous CAG repeats seen in the human disease [1]. Other studies of polylysine encoded by repeating AAA codons stall much more effectively than polylysine encoded by mixed AAG and AAA lysine codons, or just AAG codons [34, 35]. The mechanism of stalling was explained in this case by contributions from both RNA and protein [35].

Previously it was shown that inclusions formed in yeast constitutively expressing mHtt (72Q)-GFP had a diffusible core suggestive of a liquid-like state [16]. We did not observe any evidence of liquid-like states in early or late-formed inclusions. It remains possible that the early-formed inclusions detected by our dyes have already solidified into a gel state but remain disordered by the time we assessed them. It also remains possible that the structures observed in other studies are distinct to what we observed.

Of the proteins that were co-aggregated Sgta, Rack1 and Snu13 were more evenly enriched throughout the interior and outer shell of the HBRi inclusions suggesting they might form during the early inclusion assembly, whereas Ltn1, Hspb1 and Upf1 were more extensively enriched on the outside edge of the HBRi inclusions suggesting they are recruited after the inclusion has formed. Furthermore, Sgta, Hspb1 and Snu13 appeared to develop a greater enrichment to the outer shell of the mature inclusions, suggesting they are continuously enriched as inclusions mature. Previously it was shown that Hspb1 can form molecular condensates, which raises the possibility of a mixed phase separation process with polyQ that may explain some of the co-aggregation mechanism [36, 37]. Phase separation could also involve multiple immiscible phases embedded inside one another [38]. Thus, we can't rule out the possibility of proteins that are recruited at early stages of inclusion assembly forming a discrete phase that localizes to the periphery of the inclusion structure. Nonetheless, the overexpression of the proteins involved in ribosome quality control did not alter the aggregation propensity. This result is more consistent with them playing non rate-limiting roles if they are involved in aggregation or clearance, or indeed acting as bystanders that do not play a critical role in mediating inclusion formation but are co-aggregated.

In conclusion, our data suggests that nascent chains of mutant Httex1 emergent from the ribosome are unlikely to stall and therefore unlikely to drive inclusion formation as stalled

entities. However, given that we did see some ribosome-associated proteins co-aggregating as well as other proteins we previously identified as enriched in inclusions by proteomics, it remains possible that newly synthesized nascent Httex1 contributes to the aggregation process substoichiometrically by nucleating further association of post translated pools of Httex1. Alternatively, it is possible that aggregation of mHttex1 can co-aggregate endogenous Htt that is engaging in a physiological function of regulating ribosome translocation rates, and thereby draw translation machinery into the inclusions in trans. Both contexts are consistent with other reports of pre-existing pools of Httex1 monomer and small oligomers being quickly absconded into the inclusion once they form [39].

## Supporting information

**S1 Raw images.**
(TIF)

## Author Contributions

**Conceptualization:** Danny M. Hatters.

**Data curation:** Angelique R. Ormsby.

**Formal analysis:** Angelique R. Ormsby, Dezerae Cox, David Priest, Elizabeth Hinde, Danny M. Hatters.

**Funding acquisition:** Danny M. Hatters.

**Investigation:** Angelique R. Ormsby, Dezerae Cox, James Daly, David Priest.

**Methodology:** Angelique R. Ormsby, James Daly, David Priest, Elizabeth Hinde.

**Project administration:** Danny M. Hatters.

**Resources:** Danny M. Hatters.

**Software:** Dezerae Cox.

**Supervision:** Elizabeth Hinde, Danny M. Hatters.

**Writing – original draft:** Danny M. Hatters.

**Writing – review & editing:** Angelique R. Ormsby, Dezerae Cox, Danny M. Hatters.

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
