## [Decision Letter · Decision Letter 0]

25 May 2020

PONE-D-20-13286

Nascent mutant Huntingtin exon 1 chains do not stall on ribosomes during translation but aggregates do recruit machinery involved in ribosome quality control

PLOS ONE

Dear Dr. Hatters,

Thank you for submitting your manuscript to PLOS ONE. After careful consideration, we feel that it has merit but does not fully meet PLOS ONE’s publication criteria as it currently stands. Therefore, we invite you to submit a revised version of the manuscript that addresses the points raised during the review process.

Below you will find comments from the Academic Editor and one expert reviewer. Please make it clear in the introduction that the proposed mechanism is stalling due to nascent protein misfolding and justify the CAGCAA construct.  Please address comments related to statistical analysis and other comments related to data interpretation. In particular, I am not very convinced that the HBRi and PBRi are really separable entities based on the data in Fig. 5. 

We look forward to receiving your revised manuscript.

Kind regards,

David R Borchelt

Academic Editor

PLOS ONE

Journal Requirements:

2. Thank you for stating the following in the Acknowledgments Section of your manuscript: "This work was funded by grants and fellowships from the National Health and Medical Research Council Project to D.M.H. (APP1184166, APP1161803, APP1102059, APP1154352).

Please remove any funding-related text from the manuscript and let us know how you would like to update your Funding Statement. Currently, your Funding Statement reads as follows: "The funders had no role in study design, data collection and analysis, decision to publish, or preparation of the manuscript."

Additional Editor Comments (if provided):

The authors investigate whether during the translation of mRNA with a long poly Q coding sequence is impaired by ribosome stalling, and whether inclusions that form for expanded poly Q htt fragments show evidence of co-aggregation of proteins known to be involved in ribosome quality control. The paper is somewhat difficult to read due to the extensive use of jargon, but after some effort it is possible to understand the experimental data.

The authors conclude that mutant htt exon 1 mRNAs do not stall ribosomes, and that some ribosome quality control proteins do co-localize with the aggregates.

In my opinion, there are several issues with the paper in its current form that require revision.

1) The htt exon 1 construct used here is not a pure CAG repeat, but instead is a CAGCAA repeat. I would make it clear in the introduction that the design CAGCAA repeat to distinguish nascent polypeptide driven processes from any mRNA driven process that could occur with a pure CAG repeat.

2) In figure 1B and C, it would seem to me that the western blot data for the P2A T2A vector with 97Q htt is evidence for ribosome stalling. mCherry is missing in the lanes with the 97Q construct. It seems also diminished by 46Q. The fluorescence in C is for the P2A or P2A T2A vectors?

3) The technology to distinguish inclusions that may be more mature, or alternatively folded, is interesting, but it really makes the data difficult to follow. In Fig 3, are the examples of fluorescence HBRi or PBRi? The lines connect data that are from the same cells I believe. In such a case, it may be that paired T-test are more appropriate to compare each category. What statistics were used to analyze aggregation levels? What is the significance of Cerulean fluorescence levels? Is it inclusion size? Is there any meaning if only the brightest inclusions are diminished? Is there any meaning to differences in co-aggregation for a give protein between HBRi and PBRi?

4) Similar question as above for Figure 4.

5) What are the different color symbols Fig. 5A and B? Are these individual inclusions? If both the HBRi and PBRi are really just a continuum of the same type of aggregate, what is the value in distinguishing them and are they really different enough to merit separate analyses in Figs. 2-4? Data in Fig 5 suggest that these are structurally similar inclusions.

Reviewers' comments:

Reviewer's Responses to Questions

**Comments to the Author**

1. Is the manuscript technically sound, and do the data support the conclusions?

Reviewer #1: Yes

2. Has the statistical analysis been performed appropriately and rigorously? 

Reviewer #1: Yes

3. Have the authors made all data underlying the findings in their manuscript fully available?

Reviewer #1: Yes

4. Is the manuscript presented in an intelligible fashion and written in standard English?

Reviewer #1: Yes

5. Review Comments to the Author

Reviewer #1: In this study, Ormsby et al. showed that that expression of wild-type or mutant Httex1 does not stall on the ribosome during translation using a reporter assay, but proteins (Ltn1 and Rack1) involved in ribosome quality control were co-recruited into the inclusions. Additional data showed Hspb1, Sgta, and Snu13 proteins were enriched in the mutant Httx1 inclusions and could suppress the aggregation. These findings provide additional tools to understand the role of the ribosomal quality control machinery in Huntingtin’s diseases. Overall, this study is well designed and this is an interesting story, but a rigorous testing of the effects of mutant Httex1 on ribosome stalling is needed to understand the potential impact of this story. Specific concerns and suggestions are listed below.

Major concerns:

1. The authors’ claim that “long polyQ sequences do not lead to ribosome stalling” is not currently well-supported. In Figure 1, western blots showed expression of both GFP and mCherry was clearly reduced in the cells transfected with 46Q and 97Q P2A T2A constructs compared to P2A constructs. Are there differences in RNA levels in the cells transfected with these constructs? Additionally, expression of polyQ proteins should be determined by western blot or filter trap assay using anti-Htt or -polyQ antibodies.

2. CAA interrupted repeats were used for 25Q reporter construct, but no detailed sequences was provided for 46Q and 97Q constructs. Given that both repeat mRNAs and polyQ proteins may cause ribosome stalling, it is necessary to test if ribosomes could stall on CAA interrupted or pure CAG repeats RNAs.

3. The authors claim that “the over-expression of these proteins did not appear to influence the formation of inclusions of Httex1(97Q) (Fig 3C)” (lines 264-265). It is not clear however if Rack1 and Ltn1 have effects on the formation of inclusions. Additionally, the statistical analysis should be performed on impact of HaloTag protein expression on Httex1(97Q)-Cerulean inclusion formation in Fig. 3 and 4.

4. Previous studies showed that degradation and toxicity of polyQ proteins vary in different types of neurons, suggesting ribosome stalling and liquid-like properties of mutant Httex1 may be cell type dependent. These possibilities should be discussed.

Minor points

1. In Fig. 1, B and C should be clearly separated. Are western blots of P2A T2A constructs part of Fig.1C?

2. Fig. 1C should be added at the end of the sentence line 197 on page 15.

6. PLOS authors have the option to publish the peer review history of their article (what does this mean?). If published, this will include your full peer review and any attached files.

Reviewer #1: No

---

## [Author Response · Author response to Decision Letter 0]

29 Jun 2020

Please see separate document that articulates our responses.

---

## [Decision Letter · Decision Letter 1]

15 Jul 2020

Nascent mutant Huntingtin exon 1 chains do not stall on ribosomes during translation but aggregates do recruit machinery involved in ribosome quality control and RNA

PONE-D-20-13286R1

Dear Dr. Hatters,

We’re pleased to inform you that your manuscript has been judged scientifically suitable for publication and will be formally accepted for publication once it meets all outstanding technical requirements. I have one last suggestion below that you may want to consider when you are given an opportunity to address technical requirements.

Kind regards,

David R Borchelt

Academic Editor

PLOS ONE

Additional Editor Comments (optional):

The authors may want to consider whether to modify Fig 1B to only show the data for the P2A T2A vector - moving the data for the P2A vector to a supplement. The data for the P2A vector are useful knowledge, but not entirely necessary to be shown in the primary figures. It is my impression that panel C of Fig 1 is quantification of the data for the P2A T2A vector. It would be useful to make that more clear in the legend if my impression is accurate. You will see from comments of Reviewer 1 that he/she has a perception that there is some evidence for ribosome stalling in Fig. 1C, but in my opinion, the effect size is very small and your wording of very modest is acceptable.

Reviewers' comments:

Reviewer's Responses to Questions

**Comments to the Author**

1. If the authors have adequately addressed your comments raised in a previous round of review and you feel that this manuscript is now acceptable for publication, you may indicate that here to bypass the “Comments to the Author” section, enter your conflict of interest statement in the “Confidential to Editor” section, and submit your "Accept" recommendation.

Reviewer #1: (No Response)

2. Is the manuscript technically sound, and do the data support the conclusions?

Reviewer #1: Yes

3. Has the statistical analysis been performed appropriately and rigorously? 

Reviewer #1: Yes

4. Have the authors made all data underlying the findings in their manuscript fully available?

Reviewer #1: Yes

5. Is the manuscript presented in an intelligible fashion and written in standard English?

Reviewer #1: Yes

6. Review Comments to the Author

Reviewer #1: The revised manuscript has significantly been improved. For this reviewer, there are still some concerns about the claim “mHttex1 did not appear to stall translation of its own nascent chain, or at best was marginal’. In Fig. 1B, the density of mCherry bands in 97Q was clearly reduced compared to 25Q and Fig. 1C also showed statistically significant differences between 25Q and 97Q. It would be better to soften this claim based on these data.

7. PLOS authors have the option to publish the peer review history of their article (what does this mean?). If published, this will include your full peer review and any attached files.

Reviewer #1: No

---

## [Editor Report · Acceptance letter]

20 Jul 2020

PONE-D-20-13286R1 

Nascent mutant Huntingtin exon 1 chains do not stall on ribosomes during translation but aggregates do recruit machinery involved in ribosome quality control and RNA 

Dear Dr. Hatters:

I'm pleased to inform you that your manuscript has been deemed suitable for publication in PLOS ONE. Congratulations! Your manuscript is now with our production department. 

Kind regards, 

on behalf of

Prof. David R Borchelt 

Academic Editor

PLOS ONE